# Efficient Photocatalytic Degradation of Tetracycline on the MnFe_2_O_4_/BGA Composite under Visible Light

**DOI:** 10.3390/ijms24119378

**Published:** 2023-05-27

**Authors:** Xiaoyu Jiang, Qin Zhou, Yongfu Lian

**Affiliations:** Key Laboratory of Functional Inorganic Material Chemistry, Ministry of Education, School of Chemistry and Materials Science, Heilongjiang University, Harbin 150080, China; 2201357@s.hlju.edu.cn (X.J.); zhouqin@hlju.edu.cn (Q.Z.)

**Keywords:** tetracycline degradation, boron-doped graphene aerogel, manganese ferrite, peroxymonosulfate, cooperative effect

## Abstract

In this work, the MnFe_2_O_4_/BGA (boron-doped graphene aerogel) composite prepared via the solvothermal method is applied as a photocatalyst to the degradation of tetracycline in the presence of peroxymonosulfate. The composite’s phase composition, morphology, valence state of elements, defect and pore structure were analyzed by XRD, SEM/TEM, XPS, Raman scattering and N_2_ adsorption–desorption isotherms, respectively. Under the radiation of visible light, the experimental parameters, including the ratio of BGA to MnFe_2_O_4_, the dosages of MnFe_2_O_4_/BGA and PMS, and the initial pH and tetracycline concentration were optimized in line with the degradation of tetracycline. Under the optimized conditions, the degradation rate of tetracycline reached 92.15% within 60 min, whereas the degradation rate constant on MnFe_2_O_4_/BGA remained 4.1 × 10^−2^ min^−1^, which was 1.93 and 1.56 times of those on BGA and MnFe_2_O_4_, respectively. The largely enhanced photocatalytic activity of the MnFe_2_O_4_/BGA composite over MnFe_2_O_4_ and BGA could be ascribed to the formation of type I heterojunction on the interfaces of BGA and MnFe_2_O_4_, which leads to the efficient transfer and separation of photogenerated charge carriers. Transient photocurrent response and electrochemical impedance spectroscopy tests offered solid support to this assumption. In line with the active species trapping experiments, SO_4_^•−^ and O_2_^•−^ radicals are confirmed to play crucial roles in the rapid and efficient degradation of tetracycline, and accordingly, a photodegradation mechanism for the degradation of tetracycline on MnFe_2_O_4_/BGA is proposed.

## 1. Introduction

As a therapeutic drug for humans and livestock, tetracycline (TC) is largely utilized to prevent disease and promote the growth of livestock. With the widespread application of tetracycline, severe environmental pollution occurs when a large amount of tetracycline is discharged with the metabolism of human beings and animals and with industrial sewage into the environment [1,2]. Therefore, it is imperative to develop efficient techniques to eliminate TC pollution. Currently, TC sewage can be treated by biological, physical, and chemical technology [3]. However, since TC has a wide range of antibacterial effects, it is difficult to eliminate its biological toxicity through the traditional biological degradation method. Moreover, the biological method is limited by a long treatment cycle and large space for a number of bulky facilities. As for the physical treatment method, it often leads to the incomplete degradation of TC. In contrast, chemical oxidation is proven to be facile, practical, and efficient for the degradation of TC because of its easy operation, low cost, high efficiency, etc. [4].

In recent years, advanced oxidation process (AOP) as a standard chemical technology has widely been applied in the control of water pollution [5], in which hydroxyl radical (•OH) and sulfate radical (SO_4_^•−^) play a role of strong oxidants to mineralize organic pollutants. In comparison with •OH (1.7–2.7 V), SO_4_^•−^ has the advantages of high oxidation potential (2.5–3.1 V) [6], a wide range of pH applications, and a long half-life. Therefore, SO_4_^•−^ is more conducive than •OH to the rapid and efficient degradation of organic pollutants. Normally, SO_4_^•−^ generates when peroxymonosulfate (PMS) is activated by electricity [7], heat [8], ultraviolet [9], ultrasonic [10], carbon materials [11], transition metals, and/or their compounds [12,13], in which the peroxide bond –O–O– in PMS breaks to form SO_4_^•−^ and •OH [14,15].

Because of their relatively narrow band gap (1.5–2.5 eV), spinel ferrites are proper candidates for photocatalysts applicable in advanced oxidation processes [16,17]. In particular, MnFe_2_O_4_ has been evidenced to be an effective photocatalyst for the degradation of various organic pollutants by virtue of its facile synthesis, high chemical activity, excellent stability, magnetic separation, and environmental friendliness [4,17,18]. On the other hand, large-scale application of MnFe_2_O_4_ in the field of organic pollutants degradation has been limited for its low conductivity, easy aggregation, poor adsorption performance, and quick combination of photogenerated electron(e^−^)-hole(h^+^) pairs [19,20]. To date, researchers have made considerable progress in breaking these limitations through the construction of the composites of MnFe_2_O_4_ nanoparticles and low-dimensional nanomaterials. Zhou et al. synthesized the MnFe_2_O_4_/β-CD (Beta cyclodextrin) composite, which could effectively activate PMS, and displayed quite good photocatalytic degradation performance towards 2,4-dichlorophenol in a broad pH range [21]. The MnFe_2_O_4_/CeO_2_ composite synthesized by Okla et al. showed a large decrease in the recombination rate of photogenerated electron(e^−^)-hole(h^+^) pairs, leading methylene blue dye to be efficiently degraded [22]. He et al. applied the SrCoO_3_/MnFe_2_O_4_/MoS_2_ composite to the photocatalytic degradation of levofloxacin and confirmed the synergistic effect established between PMS and visible light [23].

Aiming at accelerating the transfer and separation of photogenerated carriers, graphene is often composited with photocatalysts to reduce the recombination rate of electron-hole pairs. It was reported that the MnFe_2_O_4_/rGO composite was superior to MnFe_2_O_4_ as a photocatalyst for the degradation of methylene blue under a UV lamp [4] and that PMS could be effectively activated by graphene for the degradation of methyl violet, methyl orange, and methylene blue [24]. In comparison with graphene nanosheets, 3D graphene aerogel (GA) has the advantages of strong adsorption capacity and high specific surface area, which makes it an ideal material for preparing some composite photocatalysts. In the composites of Ag_3_PO_4_/GA [25], TiO_2_/RGA [26], and TiO_2_/MoS_2_/GA [27], the semiconducting photocatalysts are uniformly dispersed among the pores of GA, improving not only their photocatalytic performance but also their stability and sustainability. Moreover, the electronic structure of GA could be easily adjusted by the covalent modification of some other non-metal elements, and a synergistic effect between the doped GA and the semiconducting photocatalyst often appears in the process of photocatalysis. For example, Chanez et al. [28] reported the enhancement of visible-light photocatalytic pollutant degradation and hydrogen evolution with a 3D nitrogen-doped GA (N-GA), and Ren et al. [29] evidenced the rapid degradation of tetracycline on the spinel CoFe_2_O_4_ nanoparticles anchored on N-GA. On the other hand, the BGA displayed a superior efficiency to GA towards the degradation of acridine orange under visible light irradiation [30], and we reported previously that a strong synergetic effect established between MnFe_2_O_4_ and BGA during the degradation of rhodamine B on the MnFe_2_O_4_/BGA [31].

In this context, the magnetic MnFe_2_O_4_/BGA composite photocatalyst was synthesized and applied to activate PMS for the efficient removal of TC from its aqueous medium under visible light. Experimental results confirmed that the as-prepared MnFe_2_O_4_/BGA composite showed significantly enhanced photocatalytic activity than that of both MnFe_2_O_4_ nanoparticles and BGA sheets for the degradation of TC. The excellent photocatalytic performance of the MnFe_2_O_4_/BGA composite could be ascribed to the type I heterojunction created on the 0D-2D point contact interface between MnFe_2_O_4_ nanoparticles and BGA sheets, which led to the broad and strong visible light absorption and the high-efficient photogenerated carrier separation.

## 2. Results and Discussion

### 2.1. Materials Characterizations

The morphology of BGA, MnFe_2_O_4_, and MnFe_2_O_4_/BGA was observed by SEM and TEM. It can be seen from Figure 1a that the irregular MnFe_2_O_4_ nanoparticles aggregate together, which is a limit to their catalytic performance. On the other hand, porous BGA is clearly observed in Figure 1b, whose average pore size is about 2.5 μm. It is no doubt that the pores of BGA are large enough to host MnFe_2_O_4_ nanoparticles. As observed in Figure 1c, the MnFe_2_O_4_ nanoparticles are evenly dispersed among BGA sheets, which is beneficial to the decrease in the agglomeration state and the increase in the specific surface area of MnFe_2_O_4_ nanoparticles. The TEM image shown in Figure 1d confirms that MnFe_2_O_4_ nanoparticles are closely wrapped by graphene sheets, which makes the MnFe_2_O_4_ nanoparticles well dispersed. On the other hand, the BGA sheets form a conductive framework in the MnFe_2_O_4_/BGA composite for the photogenerated carriers, which is also helpful for the efficient separation of the photogenerated electrons and holes.

Shown in Figure 2a are the XRD patterns of powdered MnFe_2_O_4_, BGA, and MnFe_2_O_4_/BGA. For MnFe_2_O_4_, the sharp peaks observed at 8.18°, 29.82°, 35.17°, 42.60°, 52.83°, 56.41°, and 61.76° could be assigned to the (111), (220), (311), (400), (422), (511), and (440) lattice planes of spinel MnFe_2_O_4_ (JCPDS 10-0319), respectively. Accordingly, highly pure and crystalline MnFe_2_O_4_ is prepared by the co-precipitation method. Whereas BGA demonstrates two broad peaks at 24.3° and 43°, which are ascribed separately to the (002) and (100) lattice planes of BGA. It is worth noting that both characteristic peaks of spinel MnFe_2_O_4_ and BGA are observed in the XRD pattern of MnFe_2_O_4_/BGA, indicating that spinel MnFe_2_O_4_ particles are successfully composited with BGA sheets.

As displayed in Figure 2b, MnFe_2_O_4_ demonstrates a characteristic Raman scattering peak at 616 cm^−1^ [32], and the D and G bands of BGA are separately observed at 1358 and 1589 cm^−1^. Of interest to note is that all of these Raman scattering features clearly appear in the MnFe_2_O_4_/BGA composite, offering another support to the feasible preparation of the MnFe_2_O_4_/BGA composite. It is well known that the D and G bands of carbon nanomaterials are associated with the disordered carbons defects in graphene lattice and the vibration of the sp^2^ hybridized carbon atoms in graphite, respectively. Usually, the intensity ratio of D-band to G-band (I_D_/I_G_) is applied to estimate the defects of carbon nanomaterials. For example, it can be seen from Figure 2b that the I_D_/I_G_ of MnFe_2_O_4_/BGA (0.95) is higher than that of BGA (0.86), indicating that the former has more carbon defects than the latter [24]. These extra defects might be brought about by the mutual interaction between BGA and MnFe_2_O_4_ nanoparticles, which not only prevent the magnetic MnFe_2_O_4_ nanoparticles from aggregation but also act as active sites for the photocatalytic degradation of the TC.

XPS was applied to investigate the chemical states of the elements in the MnFe_2_O_4_/BGA composite. The survey XPS spectrum shown in Figure 3a confirms the elemental composition of the MnFe_2_O_4_/BGA composite, i.e., Fe, Mn, B, C, and O. In the deconvoluted Fe 2p spectrum (Figure 3b), the peaks located at 725.4 and 711.1 eV are ascribed to Fe 2p^1/2^ and Fe 2p^3/2^, respectively, along with two satellite ones at 733.4 and 719.7 eV [6], evidencing the presence of Fe^3+^. In the deconvoluted Mn 2p spectrum (Figure 3c), the peaks located at 653.3 and 641.5 eV are assigned to Mn 2p^1/2^ and Mn 2p^3/2^, respectively, along with satellite ones at 644.9 eV, proving that the chemical state of Mn is Mn^2+^ [23]. The deconvoluted B 1s spectrum (Figure 3d) could be fitted by the two subpeaks centered at 192.4 and 193.7 eV, which are ascribed separately to BC_3_ and BC_2_O/BCO_2_ [33,34]. The deconvoluted C 1s spectrum (Figure 3e) can be divided and fitted into four subpeaks centered at 284.0, 284.5, 285.4, and 288.8eV, corresponding to C-B [35], C-C, C-O/C-O-B, and O-C=O/C-B-O [35,36] respectively. The deconvoluted O 1s spectrum (Figure 3f) is fitted by three subpeaks centered at 530.0, 531.8, and 533.3 eV, assignable to M-O [36], C-O/C-O-B, and C=O [37] bands, respectively. The above XPS spectra confirm the chemical states of every element in the MnFe_2_O_4_/BGA composite. Particularly, B atoms are doped into the frame of graphene.

In their N_2_ adsorption–desorption isotherms shown in Figure 4a, BGA, MnFe_2_O_4_, and MnFe_2_O_4_/BGA demonstrate typical IV isotherms with a type H3 hysteresis loop, indicative of the random distribution of pores and the interconnection of pores. Moreover, the calculated specific surface area of MnFe_2_O_4_/BGA (142.425 m^2^ g^−1^) is much larger than that of BGA (108.874 m^2^ g^−1^) or pure MnFe_2_O_4_ (79.068 m^2^ g^−1^). Barrett–Joyner–Halenda (BJH) analyses shown in Figure 4b reveal that the MnFe_2_O_4_/BGA composite is of mesoporous structure with pore size mainly distributed around 15.66 nm and total pore volume of 0.594 cm^3^ g^−1^. In comparison with that of BGA, the average pore size of MnFe_2_O_4_/BGA is decreased to some extent, which might be a result of the interactions between MnFe_2_O_4_ nanoparticles and BGA sheets. The large specific surface area of MnFe_2_O_4_/BGA might promote the adsorption of contaminants and offer more reactive sites to accelerate the generation of free radicals [6].

The magnetic behavior of MnFe_2_O_4_ and MnFe_2_O_4_/BGA was analyzed by a magnetic test system (VSM) at room temperature. As shown in Figure 5a, the saturation magnetization of MnFe_2_O_4_ is 76.45 emu g^−1^, and that of MnFe_2_O_4_/BGA is 8.96 emu g^−1^. Even though the saturation magnetization of MnFe_2_O_4_/BGA is largely decreased in comparison with that of MnFe_2_O_4_, it is still large enough to guarantee the effective separation of the photocatalyst from the liquid reaction system by an applied magnetic field (see the inset of Figure 5a). Thus, the MnFe_2_O_4_/BGA composite photocatalyst is reusable [17]. From an economic point of view, the recyclability of the catalyst is an important consideration for its practical applications.

The thermal stability and quantitative composition of the MnFe_2_O_4_/BGA composite were analyzed by TGA. It can be seen from Figure 5b that the MnFe_2_O_4_/BGA composite demonstrates a weight loss of 4.60 wt% between 30 and 290 °C, which is due to the evaporation of physical and chemical adsorbed moisture. The major weight loss of 60.50 wt% happens between 290 and 500 °C and is attributed to the combustion of BGA. The residues after 500 °C are ascribed to MnFe_2_O_4_ nanoparticles, which are stable till the highest temperature of TGA measurement (800 °C). On the one hand, the relative content of MnFe_2_O_4_ nanoparticles to BGA is basically consistent with that before the solvothermal process. On the other hand, the MnFe_2_O_4_/BGA composite is thermally stable to the temperature of 290 °C, satisfying the needs for practical applications.

The optical absorption property of photocatalysts is a crucial factor in determining their light-harvesting capacity. Shown in Figure 5c are the UV-visible-diffuse reflectance spectra of MnFe_2_O_4_ and MnFe_2_O_4_/BGA. It is obvious that the MnFe_2_O_4_/BGA composite has much higher absorbance than the MnFe_2_O_4_ nanoparticles in the UV-vis-NIR range. According to the Kubelka–Munk model, the direct band gaps of semiconducting materials could be estimated by the plot of (*αhν*)^2^ versus (*hν*), where α stands for extinction coefficient, and *hν* is photon energy. As shown in Fig 5d, the band gaps of MnFe_2_O_4_ and MnFe_2_O_4_/BGA were estimated to be 2.04 and 1.65 eV, respectively. Thus, with the decrease in its band gap, the MnFe_2_O_4_/BGA composite would display much-enhanced absorption capacity towards visible light, which is favorable for the generation of photon-generated carriers under visible light irradiation [17].

### 2.2. Catalytic Performance

The experimental parameters, including the doping effect of BGA, MnFe_2_O_4_/BGA dosage, PMS dosage, initial pH, and initial TC concentration, were optimized. Shown in Figure 6a is a calibration curve of TC standard solutions, which displays the mass concentration dependence of the absorbance of TC measured at the maximum absorption wavelength (λ_max_ = 357 nm). Accordingly, the concentration of TC in the reaction system was determined by its optical absorbance.

Experiments about the doping effect of BGA on the catalytic performance of MnFe_2_O_4_/BGA were conducted with the initial TC concentration, PMS dosage, composite catalyst dosage, and pH value of 20 mg L^−1^, 0.1 mmol L^−1^, 0.20 g L^−1^, and 6, respectively. As shown in Figure 6b, with the change of n from 0.1 to 1.0, the removal rate of TC is initially increased and then decreased within 60 min, and the highest removal rate reaches 67.58% when *n* = 0.5. With the increase in the load of MnFe_2_O_4_ nanoparticles on BGA sheets, more photogenerated charge carriers occur in the catalytic system. However, if BGA sheets are covered with too many MnFe_2_O_4_ nanoparticles, the light absorption of the composite catalyst would be hindered because of the shortage of efficient specific surface of BGA to light, which limits the generation of photogenerated carriers and then reduces the photocatalytic activity of the composite catalyst.

Experiments about the MnFe_2_O_4_/BGA-0.5 dosage on the catalytic performance of MnFe_2_O_4_/BGA were carried out with the initial TC concentration, PMS dosage and pH value of 20 mg L^−1^, 0.1 mmol L^−1^, and 6, respectively. It can be seen from Figure 6c that the removal rate of TC increases monotonously with the enlargement of MnFe_2_O_4_/BGA dosage. When the dosage of MnFe_2_O_4_/BGA increase gradually, more and more SO_4_^•−^ could be generated, which would contribute to more and more active sites for the degradation of TC. Of interest to note is that when the MnFe_2_O_4_/BGA dosages are 0.2 and 0.3 g L^−1^, the removal rates of TC reach 92.15% and 95.10% within 60 min, respectively. On the other hand, much visible light would be scattered or reflected if too many MnFe_2_O_4_/BGA composites are dispersed in the catalytic reaction system. Thus, a dosage of 0.2 g L^−1^ was adopted for the MnFe_2_O_4_/BGA composite catalyst in this work.

The effect of PMS dosage on TC degradation by MnFe_2_O_4_/BGA was investigated with an initial TC concentration of 20 mg L^−1^, MnFe_2_O_4_/BGA dosage of 0.20 g L^−1^, and pH value of 6. As displayed in Figure 6d, the removal rate of TC increases greatly when the dosage of PMS changes from 0.01 to 0.1 mmol L^−1^, which reaches 92.15% at the PMS dosage of 0.1 mmol L^−1^ within 60 min. This is because the steady-state concentration of SO_4_^•−^ in the reaction system would be enhanced with the increase in the dosage of PMS, offering more collision probability between SO_4_^•−^ radicals and TC molecules. However, the removal rate of TC only increases from 92.15% to 92.92% within 60 min when the dosage of PMS changes from 0.1 to 1.0 mmol L^−1^, which might be owing to the self-quenching reaction of SO_4_^•−^ radicals [6], the mutual quenching reaction of •OH and SO_4_^•−^ radicals and the quenching reaction of HSO_5_^−^ to SO_4_^•−^ radicals [38] (see Equations (1) and (2)). Thus, for the best utilization of the available SO_4_^•−^ radicals, the dosage of PMS was optimized to be 0.1 mmol L^−1^.
SO_4_^•−^ + SO_4_^•−^ → S_2_O_8_^2−^(1)
SO_4_^•−^ + •OH → HSO_5_^−^(2)
HSO_5_^−^ + SO_4_^•−^ → HSO_4_^−^ + SO_5_^•−^(3)

The initial pH dependence of TC degradation by MnFe_2_O_4_/BGA was checked out with an initial TC concentration of 20 mg L^−1^, PMS dosage of 0.1 mmol L^−1^, and MnFe_2_O_4_/BGA dosage of 0.20 g L^−1^ within 60 min. As can be seen from Figure 6e, the removal rate of TC gradually increases when the pH value changes from 2 to 6, and then largely decreases when the pH value changes from 6 to 10, which is owing to the reactions between H^+^/OH^−^ and reactive radicals (•OH and SO_4_^•−^). Nonetheless, the removal rate of TC maintains more than 85.00% when the pH value increases from 2 to 8, indicating that the MnFe_2_O_4_/BGA composite is applicable to TC degradation over a wide pH range.

The influence of initial TC concentration on the TC degradation by MnFe_2_O_4_/BGA was estimated with a PMS dosage of 0.1 mmol L^−1^, MnFe_2_O_4_/ BGA dosage of 0.20 g L^−1^, and pH value of 6. As shown in Figure 6f, the removal rate of TC decreases with the increase of the initial TC concentration. The higher the initial TC concentration, the more TC molecules would be adsorbed on the surface of the MnFe_2_O_4_/BGA composite, which leads to the cover of active sites and limits the generation of free radicals. When the initial TC concentration is 20 mg L^−1^, the removal rate of TC reaches as high as 92.15% after 60 min. This indicates that the MnFe_2_O_4_/BGA composite catalyst is an effective activator of PMS for the degradation of high-concentration TC.

To highlight the outstanding degradation ability of the PMS+MnFe_2_O_4_/BGA system towards TC, seven comparative experiments were carried out under the optimized conditions, i.e., PMS dosage 0.1 mmol L^−1^, catalyst dosage 0.20 g L^−1^, pH = 6, and initial TC concentration 20 mg L^−1^. It can be seen from Figure 7a that the TC removal ratios reach 5.24%,30.73%, 28.56%, 47.53%, 55.14%, 59.03%, and 92.15%, respectively, in the seven designed systems. It is obvious that PMS or MnFe_2_O_4_/BGA alone demonstrates very limited degradation capacity towards TC in water and that BGA or MnFe_2_O_4_ could not degrade TC efficiently in the presence of PMS. In contrast, the MnFe_2_O_4_/BGA shows the most effective removal of TC in the presence of PMS. It should be noted that the catalytic performance of the MnFe_2_O_4_/BGA catalyst is significantly better than that of the MnFe_2_O_4_/GA catalyst, confirming the importance of BGA. As electron-deficient centers, the doped B atoms in BGA could trap photogenerated electrons, which is beneficial for the transfer and separation of photogenerated carriers. Moreover, the electrons trapped by the B atom, as well as the structural defects caused by B doping, are the reaction sites for TC degradation [39]. In addition, BGA also plays multiple roles in the adsorption of TC, extending visible light absorption and suppressing the agglomeration of MnFe_2_O_4_ nanoparticles. As indicated in Figure 7b, among the seven comparative experiments, MnFe_2_O_4_/BGA also is of the highest rate constant (4.1 × 10^−2^ min^−1^), which is larger than the sum of that of MnFe_2_O_4_ (1.27 × 10^−2^ min^−1^) and that of BGA (9.85 × 10^−3^ min^−1^). Therefore, it is assumed that a synergistic effect is established between MnFe_2_O_4_ and BGA, which is responsible for the greatly enhanced photocatalytic activity of the MnFe_2_O_4_/BGA composite catalyst.

A comparison table of the photocatalytic degradation of TC on various catalysts in PMS advanced oxidation processes is shown in Table 1. It can be seen from Table 1 that the MnFe_2_O_4_/BGA composite displayed a degradation rate much higher than most of those reported in the literature. Co_3_O_4_/g-C_3_N_4_, for example, showed a lower degradation rate towards TC under the same experimental conditions. Even with much less amount of catalyst and PMS dosages, the MnFe_2_O_4_/BGA composite still exhibited a degradation rate compared with those of ZnFe_2_O_4_/Bi_2_O_2_CO_3_/BiOBr, CuWO_4−x_/Bi_12_O_17_C_l2_, or Bi_2_WO_6_/natural hematite. Though MoS_2_/Ag/g-C_3_N_4_, g-C_3_N_4_/FeWO_4_, and MCN/NCDs demonstrated higher degradation rates than the MnFe_2_O_4_/BGA composite, the data were collected with the participation of expensive metal, excess of PMS, and excess of both PMS and catalyst, respectively. Thus, the MnFe_2_O_4_/BGA composite achieved in this work makes it possible to eliminate tetracycline from wastewater in an economical way.

The reusability of the MnFe_2_O_4_/BGA composite catalyst was investigated by four successive cycling experiments. After every degradation of TC, MnFe_2_O_4_/BGA was collected by magnetic separation and subjected to the following recycling test. As displayed in Figure 8a, the removal ratio of TC declines from 92.15% to 74% in the second cycling experiment, indicative of a large loss of the catalyst activity. Nonetheless, from the second via the third to the fourth cycling experiments, the removal ratio of TC remains about 70%, confirming the cycle-to-cycle stability of the MnFe_2_O_4_/BGA composite catalyst. The apparent reaction rate constants displayed in Figure 8b offer direct support to the aforementioned results. Therefore, it is assumed that the loss of the catalyst activity might be owing to the mass loss and the leaching of metal ions for the MnFe_2_O_4_/BGA composite catalyst.

### 2.3. Activation Mechanism

The scavenging assay was performed to evaluate the reactive oxidation species (ROS) in the photocatalysis of TC on MnFe_2_O_4_/BGA, AgNO_3_, AO, TBA, phenol, and p-BQ were applied as scavengers for quenching e^−^, h^+^, •OH, SO_4_^•−^ and O_2_^•−^, respectively. It can be seen from Figure 9a that the main active species in the catalytic degradation of TC are O_2_^•^ and SO_4_^•−^. On the other hand, DMPO was used as a spin-trapping agent for the radicals of O_2_^•−^, •OH and SO_4_^•−^ in the reaction system. As shown in Figure 9b, the electron spins of DMPO-O_2_^•−^, DMPO-•OH, and DMPO-SO_4_^•−^ were clearly detectable, offering another support to the existence of O_2_^•−^ oxidation reaction in addition to the conventional ones of •OH and SO_4_^•−^ free radicals.

Shown in Figure 10a are the cyclic voltammetry curves of BGA, MnFe_2_O_4_ and MnFe_2_O_4_/BGA in a potential window of 0.2–0.6 V at a scanning rate of 20 mV·s^−1^. It is obvious that the box-like area of MnFe_2_O_4_/BGA is much larger than that of BGA or MnFe_2_O_4_, indicating that MnFe_2_O_4_/BGA demonstrates a more efficient redox reaction than BGA or MnFe_2_O_4_. From the Nyquist diagram of BGA, MnFe_2_O_4_, and MnFe_2_O_4_/BGA shown in Figure 10b, it can be seen that the arc radius of the Nyquist curve for MnFe_2_O_4_/BGA is smaller than that for MnFe_2_O_4_ or BGA, which indicates a more effective separation of photogenerated e^−^-h^+^ pairs and a higher efficiency of charge immigration across the electrode/electrolyte interface. Simultaneously, the photocurrent responses of MnFe_2_O_4_, BGA and MnFe_2_O_4_/BGA were also investigated by transient photocurrent tests. It can be seen from Figure 10c that the photocurrent of MnFe_2_O_4_/BGA is much larger than the sum of those for MnFe_2_O_4_ and BGA, confirming the aforementioned synergistic effect established between MnFe_2_O_4_ and BGA. Displayed in Figure 10d are the photoluminescence spectra of MnFe_2_O_4_, BGA, and MnFe_2_O_4_/BGA, which were excited by a laser with a wavelength of 250 nm. The room temperature luminescence peaks are observed at 420, 532, and 750 nm, respectively, in all the samples. The symmetrical peak at 750 nm might be owing to the third-order diffraction of the optical grating inside the monochromator. In comparison, the peaks at 420 and 532 nm correspond to the transitions associated with MnFe_2_O_4_ and/or BGA, which result from the recombination of photogenerated electron-hole pairs and that of photogenerated holes with singly ionized oxygen vacancies [57,58], respectively. It is obvious that the photoluminescence of the MnFe_2_O_4_/BGA composite material is greatly reduced relative to those of MnFe_2_O_4_ and BGA. This reduction is most likely due to the quenching effect caused by the charge transfer from MnFe_2_O_4_ nanoparticles to the BGA sheet. Such charge transfer-led fluorescence quenching is indicative of the effective promotion of charge transfer and efficient suppression of the recombination of photogenerated electron-hole pairs [59].

Based on the experimental results shown above, the photodegradation mechanism for the degradation of TC on MnFe_2_O_4_/BGA is proposed. As illustrated in Figure 11, MnFe_2_O_4_ nanoparticle and BGA sheets intimately contact each other to form type-I heterojunctions, owing to the chemical interaction between B doped in the sp^2^ networks of graphene and the oxygen on the surface of MnFe_2_O_4_ (B…O-Fe) as well as that between the unreduced oxygen in graphene and the Fe on the surface of MnFe_2_O_4_ (C-O…Fe-O) [31]. BGA and MnFe_2_O_4_ form heterojunctions, which create a built-in electric field at their interface to broaden light absorption spectral range, enhance charge separation, inhibit charge recombination and accelerate the kinetic process. The data shown in Figure 5, Figure 7 and Figure 9 offer strong support for the formation of type-I heterojunctions at the interface between BGA and MnFe_2_O_4._

When the MnFe_2_O_4_/BGA composite is irradiated by visible light, electron-hole pairs are generated from the semiconducting BGA and MnFe_2_O_4_, respectively (see Equations (4) and (5)). Because the conduction band (CB) energy level of MnFe_2_O_4_ is higher than that of BGA, the photogenerated electrons would transfer from MnFe_2_O_4_ to BGA owing to the contact electric field. On the other hand, since the valence band (VB) energy level of MnFe_2_O_4_ is lower than that of BGA, the photogenerated holes would migrate from MnFe_2_O_4_ to BGA, forming a straddled band alignment (see Equations (6) and (7)). Though both the photogenerated electrons and holes transfer from MnFe_2_O_4_ to BGA across type-I heterojunctions, the difference in the migration rates of electrons and holes is responsible for the efficient separation of the photogenerated charge carriers. Because of the unique configuration of BGA-loaded MnFe_2_O_4_ nanoparticles, the photogenerated charge carriers in BGA are exposed to the liquid phase and are effectively involved in the photocatalytic degradation of TC. Accordingly, the reactive radicals of O_2_^•−^, SO_4_^•−^, and •OH are formed via the processes illustrated in Equations (8)–(10), respectively. The photogenerated electrons react separately with PMS and dissolved oxygen in the water to produce SO_4_^•−^ and O_2_^•−^, while the photogenerated holes react with H_2_O to yield •OH. As indicated in Equation (11), the excellent photocatalytic performance of MnFe_2_O_4_/BGA towards the degradation of TC could be ascribed to the cooperative roles played by SO_4_^•−^, O_2_^•−^, h^+^, and •OH.
MnFe_2_O_4_ + *hν* → e− (MnFe_2_O_4_) + h^+^ (MnFe_2_O_4_)(4)
BGA + *hν* → e^−^ (BGA) + h^+^ (BGA)(5)
e^−^ (MnFe_2_O_4_) + BGA → MnFe_2_O_4_ + e^−^ (BGA)(6)
h^+^ (MnFe_2_O_4_) + BGA → MnFe_2_O_4_ + h^+^ (BGA)(7)
e^−^ (BGA) + O_2_ → O_2_^•−^(8)
e^−^ (BGA) + HSO_5_^−^ → SO_4_^•−^ + OH^−^(9)
h^+^ (BGA) + H_2_O → •OH + H^+^(10)
h^+^/•OH/O_2_^•−^/SO_4_^•−^ + TC → CO_2_ + H_2_O(11)

## 3. Materials and Methods

### 3.1. Materials

All of the following chemicals were of analytical grade and used without further purification: FeCl_3_·6H_2_O (Shanghai Aladdin Biochemical Technology Co., Ltd., Shanghai, China), MnCl_2_·4H_2_O (Shanghai Macklin Biochemical Co., Ltd., Shanghai, China), NaOH (Sinopharm Chemical Reagent Co., Ltd., Shanghai, China), KMnO_4_ (Tianjin Kemiou Chemical Reagent Co., Ltd., Tianjin, China), H_2_SO_4_ (18 mol L^−1^, Harbin Polytechnic Chemical Reagent Co., Ltd., Harbin, China), HCl (12 mol L^−1^, Harbin Polytechnic Chemical Reagent Co., Harbin, China), H_3_BO_3_ (Xilong Scientific Co., Ltd., Shantou, China), H_2_O_2_ (30 wt%, Sinopharm Chemical Reagent Co., Ltd., Shanghai, China), tetracycline (TC, Shanghai Macklin Biochemical Co., Ltd., Shanghai, China), Naphthol(Shanghai Macklin Biochemical Co., Ltd., Shanghai, China), 5-dimethyl-1-pyrroline-N-oxide (DMPO Shanghai Macklin Biochemical Co., Ltd., Shanghai, China), tertiary butyl alcohol (TBA, Shanghai Macklin Biochemical Co., Ltd., Shanghai, China), ammonium oxalate (AO, Shanghai Macklin Biochemical Co., Ltd., Shanghai, China), silver nitrate (AgNO_3_, Shanghai Macklin Biochemical Co., Ltd., Shanghai, China), p-benzoquinone (p-BQ, Shanghai Macklin Biochemical Co., Ltd., Shanghai, China), potassium monopersulfate triple salt (PMS, 42~46% KHSO_5_ basis, Shanghai Macklin Biochemical Co., Ltd., Shanghai, China), ethanol (Tianjin Zhiyuan Chemical Reagent Co., Ltd., Tianjin, China), and phenol(Shanghai Macklin Biochemical Co., Ltd., Shanghai, China). In addition, spectroscopically pure graphite powder (1000 mesh, Shanghai Xili Carbon Co. Ltd., Shanghai, China) and deionized water were applied throughout the experiments.

### 3.2. Preparation of the MnFe_2_O_4_/BGA Composite

As illustrated in Figure 12, MnFe_2_O_4_ was synthesized by the co-precipitation of MnCl_2_·4H_2_O and FeCl_3_·6H_2_O with a molar ratio of 1:2 at pH = 12 [4], whereas graphene oxide (GO) was prepared from graphite powder by modified Hummers method [60]. The MnFe_2_O_4_/BGA composite was achieved through a hydrothermal procedure. A mixture of 140 mg GO, a certain amount of MnFe_2_O_4_ (the mass ratios of MnFe_2_O_4_/GO are 0.1, 0.3, 0.5, 0.7 and 1.0, respectively), 50 mL deionized water, 20 mL ethylene glycol, and 280 mg H_3_BO_3_ were mechanically stirred for 30 min and then transferred to a Teflon liner stainless steel autoclave. After hydrothermal treatment at 180 °C for 12 h, the product was immersed in an aqueous solution of ethanol (10%) for 12 h, followed in water for another 12 h, and finally subjected to a freeze-dried treatment for 48 h. Such obtained samples were named MnFe_2_O_4_/BGA-n (*n* = 0.1, 0.3, 0.5, 0.7, or 1.0) according to the mass ratio of MnFe_2_O_4_/GO. In this context, MnFe_2_O_4_/BGA refers to MnFe_2_O_4_/BGA-0.5 without a specific statement. The MnFe_2_O_4_/GA composite and BGA were synthesized by the same procedure without the addition of H_3_BO_3_ or MnFe_2_O_4_, respectively.

### 3.3. Degradation of TC on the MnFe_2_O_4_/BGA Composite

The catalytic activity of the MnFe_2_O_4_/BGA composite catalyst was evaluated by photocatalytic degradation of TC. First, a certain amount of catalyst was added to 100 mL of 20 mg L^−1^ tetracycline solution. Then, after adjustment of the initial pH with 0.1 mol L^−1^ NaOH or 0.1 mol L^−1^ H_2_SO_4_ solutions, the reaction mixture was mechanically stirred for 10 min to ensure that the adsorption equilibrium of TC on the composite catalyst was reached. When a certain amount of PMS solution was injected, the reaction mixture was irradiated by a 300 W xenon lamp coupled with a 400 nm cut-off filter. At every ten minutes, 3 mL of the reaction mixture was taken out and filtered with 0.22 μm membrane. The concentration of residual TC in the filtrate was determined by its optical absorbance at 357 nm, and the removal rate of TC was calculated by the following formula.
(12)$$\mathrm{Degradation}=(1\;-\;\frac{C}{C_{0}})\;\times \;100\%$$
where *C*_0_ is the initial concentration of TC, and *C* is the residual concentration of TC at a specific sampling time.

The kinetic rate of TC degradation on the MnFe_2_O_4_/BGA composite catalyst was calculated in line with general pseudo-first-order kinetics.
(13)$$-\ln\frac{C}{C_{0}}\;=\;\mathrm{kt}$$
where *C*_0_ is the initial concentration of TC, *C* is the residual concentration of TC at reaction time (*t*), and *k* is the pseudo-first-order rate constant.

### 3.4. Characterization Methods

Investigation on the phase and element compositions was carried out on an X-ray diffractometer (XRD, D8 Advance, Bruker, Berlin, Germany) and an X-ray photoelectron spectroscopy (XPS, KRATOS, Stratford, UK), respectively. In comparison, morphology observation was conducted on a scanning electron microscopy (SEM, QUANTA 200S, FEI, Eindhoven, The Netherlands) and a transmission electron microscope (TEM, JEM2100, JEOL, Tokyo, Japan). Raman scattering data were collected by a Jobin Yvon (Palaiseau, France) HR 800 micro-Raman spectrometer with 458 nm excitation. Fourier transforms infrared spectra (FT-IR) were recorded by a Perkin Elmer (Waltham, MA, USA) Spectrum 100FT-IR spectrometer with background correction by referring to KBr pellets. Thermogravimetric analysis differential thermal analysis (TG-DTA) was performed at a heating rate of 10 °C min^−1^ in the air with an EXSTAR TG/DTA 7300 thermogravimetric analyzer (Hitachi, Tokyo, Japan). Brunauer-Emmett-Teller (BET) surface area collected by N_2_ adsorption–desorption method at 77 K on an ASAP 2010M analyzer (Micromeritics, Norcross, GA, USA). The magnetic behavior of MnFe_2_O_4_/BGA was recorded with an MPMS-3 superconducting quantum interference device (SQUID) magnetometer (Quantum Design, San Diego, CA, USA) in the applied field range of ±20 kOe at room temperature. The UV–vis diffuse reflectance spectra (DRS) were recorded with a UV-2550 spectrophotometer (Shimadzu, Tokyo, Japan). The optical absorbance of the TC solution was measured by a UV-3600 spectrophotometer (Shimadzu, Tokyo, Japan). Electron spin resonance (ESR) spectra were recorded with a Bruker N500 ESR spectrometer (Bruker, Rheinstetten, Germany). Cyclic voltammetry (CV), electrochemical impedance spectroscopy (EIS) and the transient photocurrent-time curves (I-t) were conducted on an SP-300 electrochemical workstation (Bio-Logic, Seyssinet-Pariset, France) in a three-electrode system with a sheet of platinum, Ag/AgCl electrode, and 0.5 mol L^−1^ Na_2_SO_4_ as a counter electrode, reference electrode and electrolyte, respectively. The working electrode is an FTO plate (1 cm^2^) homogeneously pasted with 20 mg of the composite catalyst. Photoluminescence spectra were recorded by the Edinburgh FLS980 fluorescence and phosphorescence spectrometer.

## 4. Conclusions

In summary, the MnFe_2_O_4_/BGA composite has been prepared by hydrothermal method. Homogenous dispersion of MnFe_2_O_4_ nanoparticles and effective exfoliation of BGA sheets are realized simultaneously, leading MnFe_2_O_4_ nanoparticles anchored on BGA sheets to form intimate type-I heterojunctions owing to the chemical interactions between MnFe_2_O_4_ nanoparticles and BGA sheets. In comparison with pure MnFe_2_O_4_ nanoparticles or BGA, the as-prepared MnFe_2_O_4_/BGA composite displays significantly enhanced photocatalytic activity for the degradation of high-concentrated TC under visible light irradiation, which could be mainly attributed to the type-I heterojunction induced efficient separation of photogenerated charge carriers excited from both MnFe_2_O_4_ and BGA as well as the extended visible light response. It is no doubt that this work will provide useful information for the development of a wide-spectral-responsive photocatalyst based on type I band alignment related to the narrow band gap of BGA.

Moreover, a series of AOS quenching experiments evidenced that the active radicals of SO_4_^•−^ and O_2_^•−^ played the most important roles in the effective degradation of TC on the MnFe_2_O_4_/BGA composite photocatalyst under visible light. Accordingly, a possible photodegradation mechanism for the degradation of TC was proposed, in which the multiple roles played by BGA include (a) extracting both e^−^ and h^+^ from MnFe_2_O_4_, (b) promoting the transfer and separation of photogenerated charge carriers, (c) extending visible light absorption, (d) suppressing the agglomeration of MnFe_2_O_4_ nanoparticles, (e) adsorption of TC, and (f) active sites for the degradation of TC, etc.

## Figures and Tables

**Figure 1 ijms-24-09378-f001:**
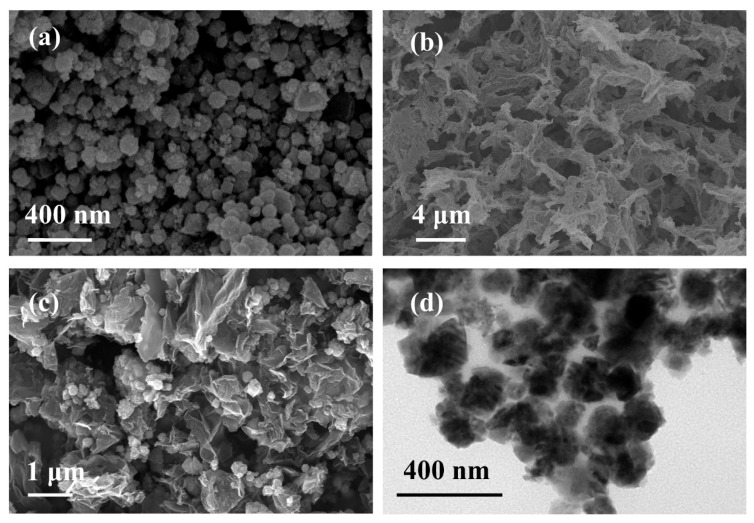
(**a**–**c**) SEM images of MnFe_2_O_4_, BGA, and MnFe_2_O_4_/BGA, respectively, and (**d**) a TEM image of MnFe_2_O_4_/BGA.

**Figure 2 ijms-24-09378-f002:**
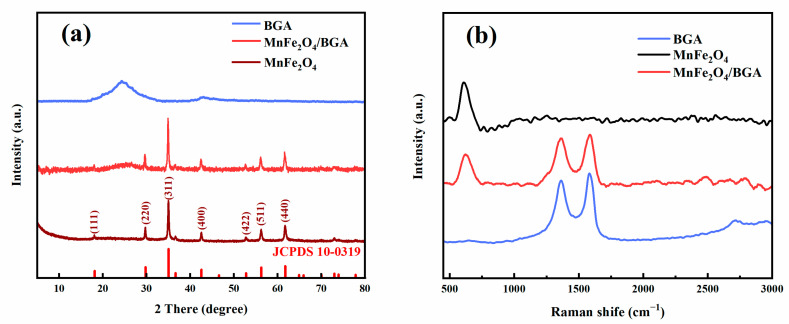
The (**a**) XRD patterns and (**b**) Raman spectra of MnFe_2_O_4_, BGA and MnFe_2_O_4_/BGA.

**Figure 3 ijms-24-09378-f003:**
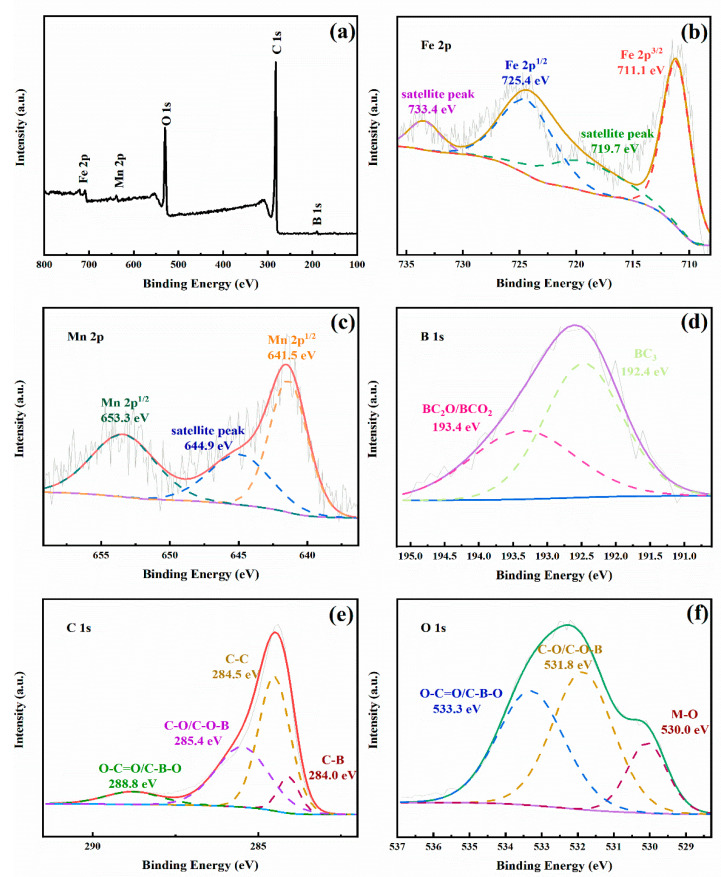
(**a**) The survey and deconvoluted (**b**–**f**) Fe 2p, Mn 2p, B 1s, C 1s, and O 1s XPS spectra of the MnFe_2_O_4_/BGA composite.

**Figure 4 ijms-24-09378-f004:**
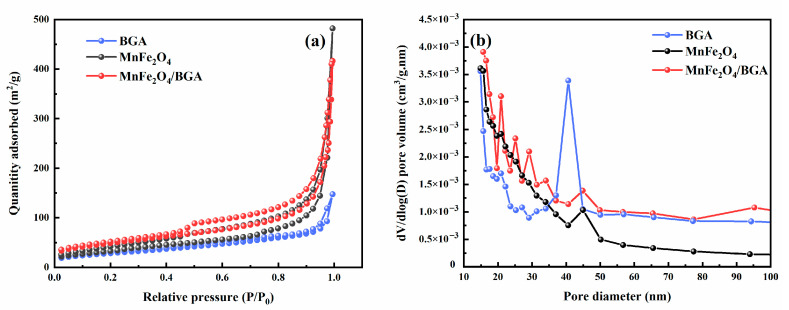
(**a**) N_2_ adsorption–desorption isotherm and (**b**) BJH pore size distribution of MnFe_2_O_4_, BGA, and MnFe_2_O_4_/BGA.

**Figure 5 ijms-24-09378-f005:**
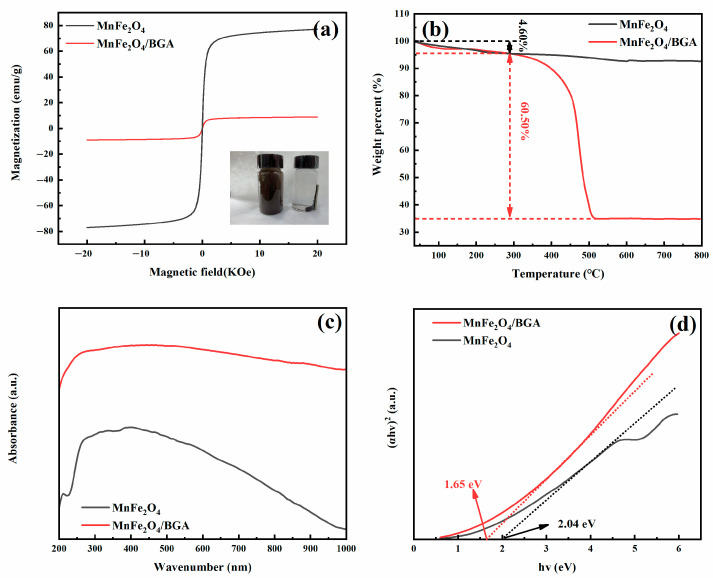
(**a**) The magnetic hysteresis loop of MnFe_2_O_4_ and MnFe_2_O_4_/BGA, the picture of MnFe_2_O_4_/BGA adsorbed by a magnet; (**b**) TG curve of MnFe_2_O_4_ and MnFe_2_O_4_/BGA; (**c**) UV-Vis-diffuse spectra; and (**d**) the (*αhν*)^2^ versus *hν* curves of BGA, MnFe_2_O_4_, and MnFe_2_O_4_/BGA.

**Figure 6 ijms-24-09378-f006:**
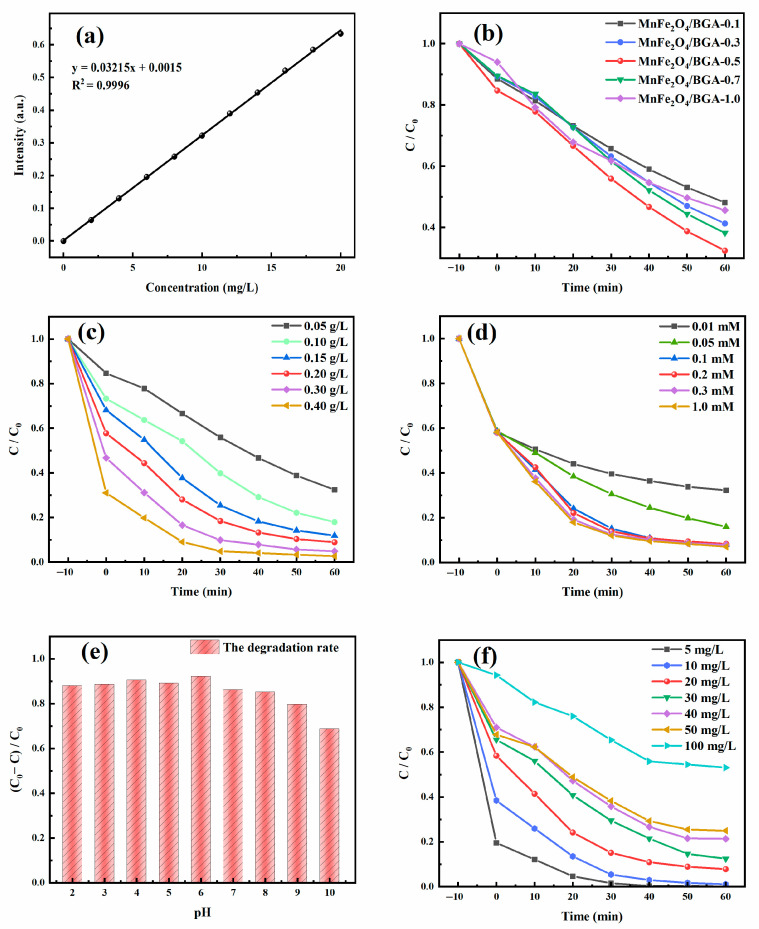
(**a**) The calibration curve of TC concentration; (**b**) MnFe_2_O_4_/BGA-n; (**c**) MnFe_2_O_4_/BGA dosage; (**d**) PMS dosage; (**e**) initial pH; and (**f**) initial concentration of TC dependences of the photocatalytic degradation of TC.

**Figure 7 ijms-24-09378-f007:**
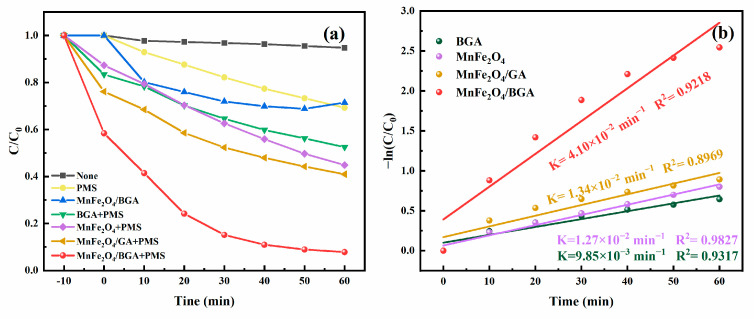
(**a**) TC removal ratios in various systems and (**b**) corresponding pseudo-first-order kinetic curves.

**Figure 8 ijms-24-09378-f008:**
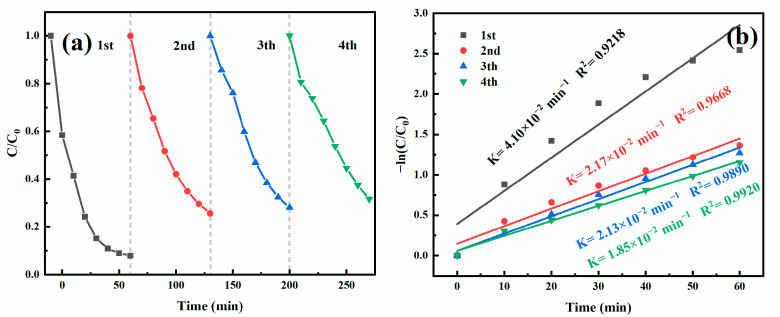
(**a**) Reusability of the MnFe_2_O_4_/BGA composite catalyst in 4 successive cycles and (**b**) corresponding pseudo-first-order kinetic curves.

**Figure 9 ijms-24-09378-f009:**
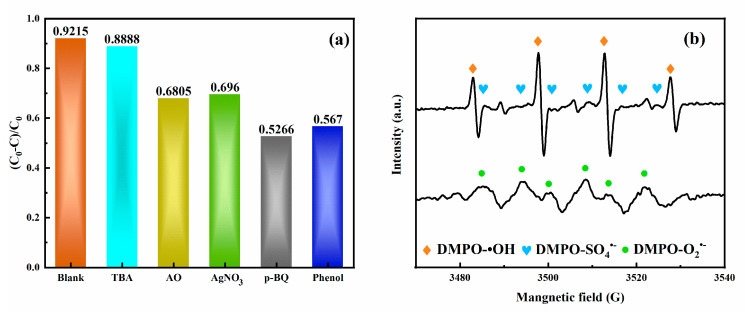
(**a**) The effect of radical scavengers on the catalytic degradation of TC and (**b**) ESR spectra of DMPO-SO_4_^•−^, DMPO-•OH, and DMPO-O_2_^•−^ detected in the system of MnFe_2_O_4_/BGA-PMS under visible light.

**Figure 10 ijms-24-09378-f010:**
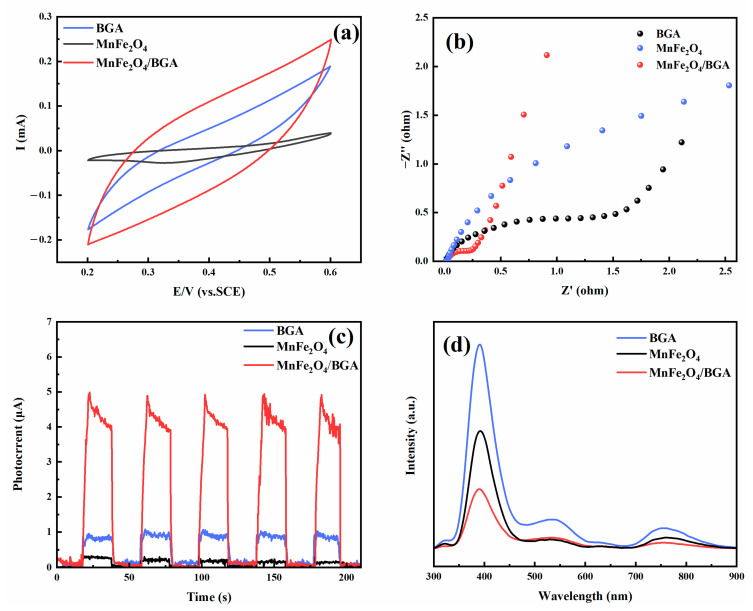
(**a**) CV curves, (**b**) EIS spectra, (**c**) I–t curves, and (**d**) photoluminescence spectra of MnFe_2_O_4_, BGA, and MnFe_2_O_4_/BGA.

**Figure 11 ijms-24-09378-f011:**
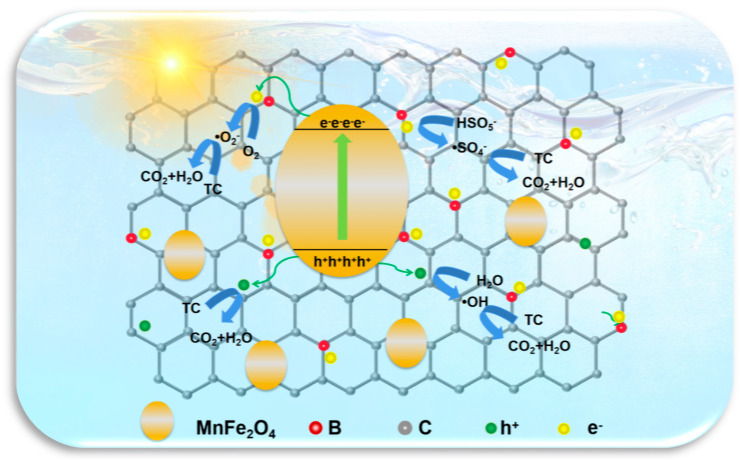
The mechanism proposed for the catalytic degradation of TC on the MnFe_2_O_4_/BGA composite.

**Figure 12 ijms-24-09378-f012:**
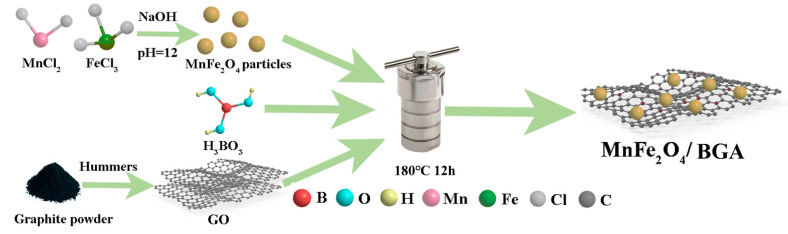
Schematic diagram of the preparation of the MnFe_2_O_4_/BGA composites.

**Table 1 ijms-24-09378-t001:** The photocatalytic degradation of TC on various catalysts in the PMS advanced oxidation process.

Catalysts	Concentration of TC (mg·L^−1^)	Catalyst Dosage (g·L^−1^)	Time (min)	Concentration of PMS (mM)	Degradation Rate (%)	Ref.
MoS_2_/Ag/g-C_3_N_4_	20	0.2	50	0.1	98.9	[40]
g-C_3_N_4_/FeWO_4_	20	0.2	60	0.6	100	[41]
ZnFe_2_O_4_/Bi_2_O_2_CO_3_/BiOBr	20	0.5	20	0.8	93.0	[42]
MCN/NCDs	10	0.5	60	0.5	98.4	[43]
CuWO_4−x_/Bi_12_O_17_C_l2_	10	0.3	30	0.2	94.7	[44]
Bi_2_WO_6_/natural hematite	50	0.5	100	0.8	91.0	[45]
FeSe_2_/Fe_3_O_4_	50	0.4	60	2.0	87.8	[46]
Zn_1−x_Cd_x_S	40	0.5	120	0.3	90.0	[47]
Mn^3+^-Co^2+^-Bi_2_O_3_	30	0.5	60	1.0	88.0	[48]
LaCoO_3_/g-C_3_N_4_	30	0.2	30	0.2	69.2	[49]
FCN-12	30	0.6	120	0.6	83.4	[50]
La-doped NiFe-LDH	20	0.04	60	2.0	90.0	[51]
Co_3_O_4_/g-C_3_N_4_	20	0.2	60	0.1	90.2	[52]
H-CoMnO_x_@NC	13	0.1	30	0.3	88.9	[53]
graphene-like biochar/g-C_3_N_4_	10	0.2	60	0.3	~90	[54]
Co_0.5_Cu_0.5_Fe_2_O_4_	10	0.06	40	0.1	86.0	[55]
CuBi_2_O_4_/BiOBr	10	0.2	35	2.0	90.3	[56]
MnFe_2_O_4_/BGA	20	0.2	60	0.1	92.2	This work

## Data Availability

Data sharing is not applicable to this article.

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
