# Peer review of "Efficient Photocatalytic Degradation of Tetracycline on the MnFe_2_O_4_/BGA Composite under Visible Light"

_ijms, 2023, doi:10.3390/ijms24119378_

Round 1
Reviewer 1 Report
This manuscript reports the photocatalytic degradation of tetracycline (TC) under visible light using the catalyst, MnFe2O4/BGA (boron-doped graphene aerogel). The MnFe2O4/BGA catalyst has demonstrated great performance for the photocatalytic degradation of TC owning to its enhanced charge separation, synergistic effects between MnFe2O4 and BGA, high porosity and surface area, visible light absorption efficiency, and ease of separation.
1. The authors mentioned in the caption of Figure 1 that Figure 1a and Figure 1b correspond to BGA and MnFe2O4, respectively. This was also re-stated in line 97 on page 3. Based on the discussion provided in the paragraph below Figure 1, it is evident that Figure 1a correspond to MnFe2O4 and Figure 1b corresponds to BGA.
2. On page 8, lines 242 to 245, the authors mentioned that because the removal rate of TC only increase from 92.15% to 92.92% within 60 minutes when the dosage of PMS changes from 0.1 to 1.0 mmol L-1 and might be attributed to the mutual quenching reaction of excessive SO4- radicals. Would you be able to justify the statement about excessive SO4- radicals with experiments?
3. Figure 6e, shows the dependence of pH on the photocatalytic degradation of TC. What reaction time was used for this study? This information will be useful in the discussion on page 9, first paragraph.
4. Based on the results presented in Figure 7, the MnFe2O4/BGA catalyst performed much better than the MnFe2O4/GA catalyst, can the author elaborate and provide additional explanation on the reason for the discrepancy in the results?
The manuscript is written relatively well, it can use minor grammar corrections and proof reading of the sentences for a slight increase in scientific soundness.
Reviewer 2 Report
The work is a rehash of "The Efficient Photocatalytic Degradation of Organic Pollutants on the MnFe2O4/BGA Composite under Visible Light" (Nanomaterials, 2021, 11). What is the novelty? Tetracycline degradation?
The manuscript has quality of english language.
Reviewer 3 Report
The work by Jiang et al. is very interesting and presents new results for the photocatalytic application of heterostructures. Furthermore, the manuscript is well written and the authors have made a systematic study of this MnFe2O4/BGA-based system. I believe that the present study can be accepted in the International Journal of Molecular Science, after further revisions. Hence, the comments on the present manuscript are given below:
1) I recommend not abbreviating the words in the abstract.
2) Regarding the PL emission from these samples (see Figure 10), it is interesting to analyze the spectrum in the entire visible region, especially to investigate the transitions associated with the MnFe2O4 phase present in the system. Furthermore, for this reviewer, this reduction in PL emission of composite material is most likely due to a quenching effect. The authors could discuss more the quenching effect on PL emission.
3) I believe these recent studies may help in this work and I recommend adding these references to this manuscript.
https://doi.org/10.1021/ie200162a
https://doi.org/10.1016/j.ceramint.2018.06.190
https://doi.org/10.1016/j.materresbull.2023.112242
https://doi.org/10.1016/j.ceramint.2016.03.019
https://doi.org/10.1007/978-3-030-62226-8_12
https://doi.org/10.1016/j.jphotochem.2023.114612
The article is well written.
Round 2
Reviewer 2 Report
The manuscript is a copy of the other work.
Reviewer 3 Report
The authors of the manuscript responded to the comments of this reviewer. However, I would like to request some additional modifications.
1) Authors must expand the abstract in a more concise.
2) Please provide a very convincing and meaningful figure in the introduction section.
3) Include a comparison table with other composite materials as photocatalysts, in order to demonstrate the satisfactory efficiency obtained for this new photocatalyst.
4) Please include a scheme for the synthesis of these materials under investigation.
Acceptable
